# Off-policy Bandits with Deficient Support

## Abstract

Off-policy training of contextual-bandit policies is attractive in online systems (e.g. search, recommendation, ad placement), since it enables the reuse of large amounts of log data. State-of-the-art methods for off-policy learning, however, are based on inverse propensity score (IPS) weighting, which requires that the logging policy chooses all actions with non-zero probability for any context (i.e., full support). In real-world systems, this condition is often violated, and we show that existing off-policy learning methods based on IPS weighting can fail catastrophically. We therefore develop new off-policy contextual-bandit methods that can controllably and robustly learn even when the logging policy has deficient support. To this effect, we explore three approaches that provide various guarantees for safe learning despite the inherent limitations of support deficient data: restricting the action space, reward extrapolation, and restricting the policy space. We analyze the statistical and computational properties of these three approaches, and empirically evaluate their effectiveness in a series of experiments. We find that controlling the policy space is both computationally efficient and that it robustly leads to accurate policies.

## 1 Introduction

Many interactive systems (e.g., voice assistants, recommender systems, ad placement) can be modeled as *contextual bandit* problems (Langford & Zhang, 2008). In particular, each user request provides a context (e.g., user profile, query) for which the system selects an action (e.g., recommended product, presented ad) and receives a reward (e.g., purchase, click). Such contextual-bandit data is logged in large quantities as a by-product of normal system operation (Li et al., 2011; 2015; Joachims et al., 2017), making it an attractive and low-cost source of training data. With terabytes of such log data readily available in many online systems, a range of algorithms have been proposed for batch learning from such logged contextual-bandit feedback (Strehl et al., 2011; Dudík et al., 2011; Swaminathan & Joachims, 2015a; Thomas & Brunskill, 2016; Farajtabar et al., 2018; Su et al., 2019; London & Sandler, 2019). However, as we will argue below, these algorithms require an assumption about the log data that makes them unsuitable for many real-world applications.

This assumption is typically referred to as the positivity or support assumption, and it is required by the Empirical Risk Minimization (ERM) objective that these algorithms optimize. Specifically, unlike in online learning for contextual bandits (Williams, 1992; Agarwal et al., 2014), batch learning from bandit feedback (BLBF) operates in the off-policy setting. During off-policy learning, the algorithm has to address the counterfactual question of how much reward each policy in the policy space would have received, if it had been used instead of the logging policy. To this effect, virtually all state-of-the-art off-policy learning methods for contextual-bandit problems rely on counterfactual estimators (Bottou et al., 2013; Dudík et al., 2011; Swaminathan & Joachims, 2015a; Thomas & Brunskill, 2016; Farajtabar et al., 2018; Su et al., 2019) that employ inverse propensity score (IPS) weighting to get an unbiased ERM objective. Unlike regression-based direct-modeling (DM) approaches that are often hampered by bias from model-misspecification, IPS allows a controllable bias-variance trade-off through clipping and other variance-regularization techniques (Strehl et al., 2011; Swaminathan & Joachims, 2015a; London & Sandler, 2019).

Unfortunately, IPS and its variance-control mechanisms break down when the logging policy does not have full support – meaning that some actions have zero probability of being selected under the logging policy. In this case IPS can be highly biased. Full support is an unreasonable assumption in many real-world systems, especially when the action space is large and many actions have poor

rewards. For example, in a recommender system with a large catalog (e.g. movies, music), it may be that less than 10% of the actions have support under the logging policy. We will show that existing learning algorithms can fail catastrophically on such support deficient data.

In this paper, we develop new off-policy contextual-bandit algorithms that are specifically designed to deal with support deficient log data. Since support deficiency translates into blind spots where we do not have any knowledge about the rewards, accounting for these blind spots as part of learning is crucial for robust learning. We approach this problem from three perspectives. First, we explore restricting the action space to those actions that have support under the logging policy. Second, we explore imputation methods that extrapolate estimated rewards to those blind spots. And, third, we restrict the policy space to only those policies that have limited exposure to the blind spots. To make the latter approach computationally tractable, we define a new measure of Support Divergence between policies, show how it can be estimated efficiently without closed-form knowledge of the logging policy, and how it can be used as a constraint on the policy space. We analyze the statistical and computational properties of all three approaches and perform an extensive empirical evaluation. We find that restricting the policy space is particularly effective, since it is computationally efficient, empirically effective at learning good policies, and convenient to use in practice.

## 2   RELATED WORK

Most prior works on BLBF can be classified into two different approaches. The first – called Direct Model (DM) – is based on a reduction to supervised learning, where a regression estimate is trained to predict rewards (Beygelzimer & Langford, 2009). To derive a policy, the action with the highest predicted reward is chosen. A drawback of this simple approach is the bias that results from mis-specification of the regression model. Since regression models are often substantially misspecified for real-world data, the DM approach often does not work well empirically.

The second approach is based on policy learning via ERM with a counterfactual risk estimator. Inverse propensity score (IPS) weighting is one of the most popular estimators to be used as empirical risk. However, policy learning algorithms based on IPS and related estimators (Strehl et al., 2011; Swaminathan & Joachims, 2015a;b; Thomas & Brunskill, 2016; London & Sandler, 2019) require the assumption that the logging policy has full support for every policy in the policy space. One exception is the work of Liu et al. (2019). They relax the assumption to the existence of an optimal policy such that the logging policy covers the support of this optimal policy. However, this is an untestable assumption that does not provide guarantees for real-world applications.

Our work proposes three approaches to addressing off-policy learning with support deficiency. First, our conservative extrapolation method is related to the method proposed by Liu et al. (2019). They focus on the correction of the state distribution by defining an augmented MDP, and pessimistic imputation is used to get an estimate for policy-gradient learning. Second, our method of restricting the policy space uses a surrogate for the support divergence of two policies that was previously used as control variate in the SNIPS estimator (Swaminathan & Joachims, 2015b). It also appeared in the Lagrangian formulation of the BanditNet objective (Joachims et al., 2018) and in the gradient update in REINFORCE algorithm (Williams, 1992). This connection gives interesting new insight that the baselines used in policy-gradient algorithms not only help to reduce variance in gradients (Greensmith et al., 2004), but that they also connect to the problem of support deficiency in the off-policy setting.

## 3   OFF-POLICY LEARNING WITH DEFICIENT SUPPORT

We start by formally defining the problem of learning a contextual-bandit policy in the BLBF setting. Input to the policy are contexts $x \in \mathcal{X}$ drawn i.i.d from a fixed but unknown distribution $P(\mathcal{X})$. Given context $x$, the system executes a possibly stochastic policy $\pi(\mathcal{Y}|x)$ that selects an action $y \in \mathcal{Y}$. For this context and action pair, the system observes a reward $r \in [r_{min}, r_{max}]$ from $P(r|x, y)$. Given a space of policies $\Pi$, the reward of any policy $\pi \in \Pi$ is defined as

$$R(\pi) = \mathbb{E}_{x} \mathbb{E}_{y \sim \pi(y|x)} \mathbb{E}_{r \sim P(r|x,y)} [r]. \tag{1}$$

In the BLBF setting, the learning algorithm is given a dataset

$$\mathcal{D} := \{x_i, y_i, r_i, \pi_0(y_i|x_i)\}_{i=1}^{n}$$

of past system interactions which consists of context-action-reward-propensity tuples. The propensity $\pi_0(y_i|x_i)$ is the probability of selecting action $y_i$ for context $x_i$ under the policy $\pi_0$ that was used to log the data. We call $\pi_0$ the logging policy, and we will discuss desired conditions on the stochasticity of $\pi_0$ in the following. The goal of off-policy learning is to exploit the information in the logged data $\mathcal{D}$ to find a policy $\hat{\pi} \in \Pi$ that has high reward $R(\hat{\pi})$.

Analogous to the ERM principle in supervised learning, off-policy learning algorithms typically optimize a counterfactual estimate $\hat{R}(\pi)$ of $R(\pi)$ as the training objective (Li et al., 2011; 2015; Bottou et al., 2013; Swaminathan & Joachims, 2015a).

$$\hat{\pi} = \arg\max_{\pi \in \Pi}[\hat{R}(\pi)] \tag{2}$$

For conciseness, we ignore additional regularization terms in the objective (Swaminathan & Joachims, 2015a), since they are irrelevant to the main point of this paper. As counterfactual estimator $\hat{R}(\pi)$, most algorithms rely on some form of IPS weighting (Strehl et al., 2011; Dudík et al., 2011; Swaminathan & Joachims, 2015a;b; Wang et al., 2017; Su et al., 2019) to correct the distribution mismatch between the logging policy $\pi_0$ and each target policy $\pi \in \Pi$.

$$\hat{R}_{IPS}(\pi) = \frac{1}{n}\sum_{i=1}^{n} \frac{\pi(y_i|x_i)}{\pi_0(y_i|x_i)} r_i. \tag{3}$$

A crucial condition for the effectiveness of the IPS estimator (and similar estimators) is that the logging policy $\pi_0$ assigns non-zero probability to all actions that have non-zero probability under the target policy $\pi$ we aim to evaluate. This condition is known as positivity or full support, and it is defined as follows.

**Definition 1** (Full support). *The logging policy $\pi_0$ is said to have full support for $\pi$ when $\pi_0(y|x) > 0$ for all actions $y \in \mathcal{Y}$ and contexts $x \in \mathcal{X}$ for which $\pi(y|x) > 0$.*

It is known that the IPS estimator is unbiased, $\mathbb{E}_{\mathcal{D}}[\hat{R}_{IPS}(\pi)] = R(\pi)$, if the logging policy $\pi_0$ has full support for $\pi$ (Li et al., 2011). To ensure unbiased ERM, algorithms that use the IPS estimator require that the logging policy $\pi_0$ has full support for all policies $\pi \in \Pi$ in the policy space. For sufficiently rich policy spaces, like deep-networks $f_w(x, y)$ with softmax outputs of the form

$$\pi_w(y|x) = \frac{exp(f_w(x, y))}{\sum_{y' \in \mathcal{Y}} exp(f_w(x, y'))}, \tag{4}$$

this means that the logging policy $\pi_0$ needs to assign non-zero probability to every action $y$ in every context $x$. This is a strong condition that is not feasible in many real-world systems, especially if the action space is large and many actions have poor reward.

If the support requirement is violated, ERM learning can fail catastrophically. We will show below that the underlying reason is bias, not excessive variance that could be remedied through clipping or variance regularization (Strehl et al., 2011; Swaminathan & Joachims, 2015a). To quantify how support deficient a logging policy is, we denote the set of unsupported actions for context $x$ under $\pi_0$ as

$$\mathcal{U}(x, \pi_0) := \{y \in \mathcal{Y} | \pi_0(y|x) = 0\}.$$

The bias of the IPS estimator is then characterized by the expected reward on the unsupported actions.

**Proposition 1.** *Given contexts $x \sim P(\mathcal{X})$ and logging policy $\pi_0(\mathcal{Y}|x)$, the bias of $\hat{R}_{IPS}$ for target policy $\pi(\mathcal{Y}|x)$ is equal to the expected reward on the unsupported action sets, i.e., $bias(\pi|\pi_0) = \mathbb{E}_x[-\sum_{y \in \mathcal{U}(x,\pi_0)} \pi(y|x)\delta(x, y)]$.*

The proof is in Appendix A.1. From Proposition 1, it is clear that support deficient log data can drastically mislead ERM learning. To quantify the effect of support deficiency on ERM, we define the support divergence between a logging policy $\pi_0$ and a target policy $\pi$ as follows.

**Definition 2** (Support Divergence). *For contexts $x \sim P(\mathcal{X})$ and any corresponding pair of target policy $\pi$ and logging policy $\pi_0$, the Support Divergence is defined as*

$$\mathcal{D}_{\mathcal{X}}(\pi|\pi_0) := \mathop{\mathbb{E}}_{x \sim P(\mathcal{X})}\left[\sum_{y \in \mathcal{U}(x,\pi_0)} \pi(y|x)\right]. \tag{5}$$

With this definition in hand, we can quantify the effect of support deficiency on ERM learning for a policy space $\Pi$ under logging policy $\pi_0$.

**Theorem 1.** *For any given hypothesis space $\Pi$ with logging policy $\pi_0 \in \Pi$, there exists a reward distribution $\mathcal{P}_r$ with support in $[r_{min}, r_{max}]$ such that in the limit of infinite training data, ERM using IPS over the logged data $\mathcal{D} \sim P(\mathcal{X}) \times \pi_0(\cdot|\mathcal{X}) \times \mathcal{P}_r$ can select a policy $\hat{\pi} \in \arg\max_{\pi \in \Pi} \mathbb{E}_{\mathcal{D}}[\hat{R}_{IPS}(\pi)]$ that is at least $(r_{max} - r_{min}) \max_{\pi \in \Pi} \mathcal{D}_{\mathcal{X}}(\pi|\pi_0)$ suboptimal.*

The proof is in Appendix A.2. To illustrate the theorem, consider a problem with rewards $r \in [-1, 0]$. Furthermore, consider a policy space $\Pi$ that contains a good policy $\pi_g$ with $R(\pi_g) = -0.1$ and a bad policy $\pi_b$ with $R(\pi_b) = -0.7$. If policy $\pi_b$ has support divergence $\mathcal{D}_{\mathcal{X}}(\pi_b|\pi_0) = 0.6$ or larger, then ERM may return the bad $\pi_b$ instead of $\pi_g$ even with infinite amounts of training data.

Note that it is sufficient to merely have one policy in $\Pi$ that has large support deficiency to achieve this suboptimality. It is therefore crucial to control the support divergence $\mathcal{D}_{\mathcal{X}}(\pi|\pi_0)$ uniformly over all $\pi \in \Pi$, or to account for the suboptimality it can induce. To this effect, we explore three approaches in the following.

### 3.1 SAFE LEARNING BY RESTRICTING THE ACTION SPACE

The first and arguably most direct approach to reducing $\mathcal{D}_{\mathcal{X}}(\pi|\pi_0)$ is to disallow any action that has zero support under the logging policy. For the remaining action set, the logging policy has full support by definition. This restriction of the action set can be achieved by transforming each policy $\pi \in \Pi$ into a new policy that sets the probability of the unsupported actions to zero.

$$\pi(y|x) \longrightarrow \bar{\pi}(y|x) := \frac{\pi(y|x)\, \mathbb{I}_{\{y \notin \mathcal{U}(x,\pi_0)\}}}{1 - \sum_{y' \in \mathcal{U}(x,\pi_0)} \pi(y'|x)} \tag{6}$$

This results in a new policy space $\bar{\Pi}$. All $\bar{\pi} \in \bar{\Pi}$ have support divergence of zero $\mathcal{D}_{\mathcal{X}}(\bar{\pi}|\pi_0) = 0$ and ERM via IPS is guaranteed to be unbiased.

While this transformation of the policy space from $\Pi$ to $\bar{\Pi}$ is conceptually straightforward, it has two potential drawbacks. First, restricting the action space without any exceptions may overly constrain the policies in $\bar{\Pi}$. In particular, if the optimal action $y^*$ for a specific context $x$ does not have support under the logging policy, no $\bar{\pi} \in \bar{\Pi}$ can ever choose $y^*$ even if there are many observations of similar $y$'s on similar context $x'$. The second drawback is computational. For every context $x$ during training and testing, the system needs to evaluate the logging policy $\pi_0(y|x)$ to compute the transformation from $\pi$ to $\bar{\pi}$. This can be prohibitively expensive especially at test time, where – after multiple rounds of off-policy learning with data from previously learned policies – we would need to evaluate the whole sequence of previous logging policies to execute the learned policy.

### 3.2 SAFE LEARNING THROUGH REWARD EXTRAPOLATION

As illustrated above, support deficiency is a problem of blind spots where we lack information about the rewards of some actions in some contexts. Instead of disallowing the unsupported actions like in the previous section, an alternative is to extrapolate the observed rewards to fill in the blind spots. To this effect, we propose the following augmented IPS estimator that imputes an extrapolated reward $\hat{\delta}(x, y)$ for each unsupported action $y \in \mathcal{U}(x, \pi_0)$.

$$\hat{R}_{IPS}^{\delta}(\pi) = \frac{1}{n} \sum_{i=1}^{n} \left[ \frac{\pi(y_i|x_i)}{\pi_0(y_i|x_i)} r_i + \sum_{y \in \mathcal{U}(x_i, \pi_0)} \pi(y|x_i)\hat{\delta}(x_i, y) \right] \tag{7}$$

In the following proposition, we characterize the bias of the augmented IPS estimator for any given reward extrapolation $\hat{\delta}(x, y)$. We denote the mean of the reward $r$ for context $x$ and action $y$ with $\delta(x, y) = \mathbb{E}_{r \sim P(r|x,y)}[r]$. Furthermore, $\Delta(x, y) := \hat{\delta}(x, y) - \delta(x, y)$ denotes the error of the reward extrapolation for each $x$ and $y$.

**Proposition 2.** *Given contexts $x_1, x_2, \ldots, x_n$ drawn i.i.d from the unknown distribution $P(\mathcal{X})$, for action $y_i$ drawn independently from logging policy $\pi_0$ with probability $\pi_0(\mathcal{Y}|x_i)$, the bias of the empirical risk defined in Equation (7) is $\mathbb{E}_x[\sum_{y \in \mathcal{U}_x^{\pi_0}} \pi(y|x)\Delta(x, y)]$.*

In this way we can learn in the original action and policy space, but mitigate the effect of the support deficiency by explicitly incorporating the extrapolated reward $\hat{\delta}(x, y)$. We explore two choices for $\hat{\delta}(x, y)$ in the following, which provide different types of guarantees.

**Conservative Extrapolation.** To minimize the user impact of randomization in the logging policy, it is generally desirable to put zero probability on actions the are very likely to have low (or even catastrophic reward). This means that precisely those bad actions are likely to not be supported in the logging policy. A key danger of blind spots regarding those actions is that naive IPS training will inadvertently learn

---

**Algorithm 1:** Data Augmentation

input: original logged dataset $\mathcal{D}$, replaycount $k$,
  reward estimate $\hat{\delta}(x, y)$; output: $\mathcal{D}'$;
initialization: $\mathcal{D}' = \emptyset$ ;
**for** $j = 1, \ldots, k$ **do**
  **for** $i = 1, \ldots, n$ **do**
    Define $U_{x_i}$ to be the uniform distribution
      over $\mathcal{U}(x_i, \pi_0)$;
    Draw $y \sim U_{x_i}$;
    $\mathcal{D}' = \mathcal{D}' \bigcup \{x_i, y, \hat{\delta}(x_i, y), \frac{1}{|\mathcal{U}(x_i, \pi_0)|}\}$;
  **end**
**end**

---

a policy that selects those actions. This can be avoided by being maximally conservative about unsupported actions and imputing the lowest possible reward $\forall x, y \in \mathcal{U}(x, \pi_0) : \hat{\delta}(x, y) = r_{min}$. Intuitively, by imposing the worst possible reward for the unsupported actions, the learning algorithm will aim to avoid these low-reward areas. However, unlike for the $\bar{\pi}$ policies resulting from the restricted action space, the learned policy is not strictly prohibited from choosing unsupported actions – it is merely made aware of the maximum loss that the action may incur. Note that for problems where $r_{min} = 0$, the naive IPS estimator is identical to conservative extrapolation since the second term in Equation (7) is zero.

**Regression Extrapolation.** Instead of extrapolating with the worst-case reward, we may have additional prior knowledge in the form of a model-based estimate that reduces the bias. In particular, we explore using a regression estimate $\hat{\delta} = \arg\min_{\hat{\delta}^\theta} \frac{1}{n} \sum_{i=1}^{n} (\hat{\delta}^\theta(x_i, y_i) - r_i)^2$ that extrapolates from the observed data $\mathcal{D}$. Typically, $\hat{\delta}^\theta$ comes from a parameterized class of regression functions. Other regression objectives could also be used, such as weighted linear regression that itself uses importance sampling as weights (Farajtabar et al., 2018). But, fundamentally, all regression approaches assume that the regression model is not misspecified and that it can thus extrapolate well. Note that the IPS part of Equation (7) can be changed to any estimators (with action set restricted on $\mathcal{U}(x, \pi_0)^c$ for all $x$), and it turns out that doubly robust (Dudík et al., 2011) and CAB (Su et al., 2019) are special extensions of regression extrapolation that substitute the IPS part with their corresponding estimator.

**Efficient Approximation.** Evaluating the augmented IPS estimator from Equation (7) can be computationally expensive if the number of unsupported actions $\mathcal{U}(x, \pi_0)$ is large. To overcome this problem, we propose to use sampling to estimate the expected reward on the unsupported action, which can be thought of as augmenting the dataset $\mathcal{D}$ with additional observations where the logging policy has zero support. In particular, we propose the data-augmentation procedure detailed in Algorithm 1. With the additional bandit data $\mathcal{D}' = \{x'_j, y'_j, \hat{\delta}(x'_j, y'_j), p'_j\}_{j=1}^{m}$ from Algorithm 1, the new objective is

$$\arg\min_{\pi \in \Pi} \left\{ \frac{1}{n} \sum_{i=1}^{n} \frac{\pi(y_i|x_i)}{\pi_0(y_i|x_i)} r_i + \frac{1}{m} \sum_{j=1}^{m} \frac{\pi(y'_j|x'_j)}{p'_j} \hat{\delta}(x'_j, y'_j) \right\}. \quad (8)$$

In Appendix A.5, we show that the empirical risk in Equation (8) has the same expectation (over randomness in $\mathcal{D}$ and $\mathcal{D}'$) as $\hat{R}^\delta_{IPS}(\mathcal{D})$ and can thus serve as an approximation for Equation (7).

## 3.3 Safe Learning by Restricting the Policy Space

As motivated by Theorem 1, the risk of learning from support deficient data scales with the maximum support divergence $\mathcal{D}_\mathcal{X}(\pi|\pi_0)$ among the policies in the policy space $\Pi$. Therefore, our third approach restricts the policy space to the subset $\Pi^\kappa \subset \Pi$ that contains the policies $\pi \in \Pi$ with an acceptably low support divergence $\mathcal{D}_\mathcal{X}(\pi|\pi_0) \leq \kappa$.

$$\Pi^\kappa = \{\pi | \pi \in \Pi \wedge \mathcal{D}_\mathcal{X}(\pi|\pi_0) \leq \kappa\} \quad (9)$$

The parameter $\kappa$ has an intuitive meaning. It specifies the maximum probability mass that a learned policy can place on unsupported actions. By limiting this to $\kappa$, we limit the maximum bias of

the ERM procedure according to Proposition 2 while not explicitly torquing the rewards like in conservative reward imputation.

A key challenge, however, is implementing this restriction of the hypothesis space, such that the ERM learner $\hat{\pi} = \arg\max_{\pi \in \Pi^\kappa}[\hat{R}_{IPS}(\pi)]$ only considers the subset $\Pi^\kappa \subset \Pi$. In particular, we do not have access to the context distribution $P(\mathcal{X})$ for calculating $\mathcal{D}_{\mathcal{X}}(\pi|\pi_0)$, nor would it be possible to enumerate all $\pi \in \Pi$ to check the condition $\mathcal{D}_{\mathcal{X}}(\pi|\pi_0) \leq \kappa$, which itself requires a possibly infeasible iteration over all actions. The following theorem (with proof in Appendix A.3) gives us an efficient way of estimating and controlling $\mathcal{D}_{\mathcal{X}}(\pi|\pi_0)$ without explicit knowledge of $P(\mathcal{X})$ or access to the logging policy $\pi_0$ beyond the logged propensities.

**Theorem 2.** *For contexts $x_i$ drawn i.i.d from $P(\mathcal{X})$, action $y_i$ drawn from logging policy $\pi_0$, we define $S_{\mathcal{D}}(\pi|\pi_0) = \frac{1}{n}\sum_{i=1}^{n} \frac{\pi(y_i|x_i)}{\pi_0(y_i|x_i)}$. For any policy $\pi$ it holds that*

$$\underset{x \sim P(\mathcal{X})}{\mathbb{E}} \underset{y \sim \pi_0(\cdot|x)}{\mathbb{E}} [S_{\mathcal{D}}(\pi|\pi_0)] + \mathcal{D}_{\mathcal{X}}(\pi|\pi_0) = 1 \tag{10}$$

Using this theorem, the following proposition (proof in Appendix A.4, empirically verified in Appendix B) gives us an efficient way of implementing the constraint $\mathcal{D}_{\mathcal{X}}(\pi|\pi_0) \leq \kappa$ via $1 - S_{\mathcal{D}}(\pi|\pi_0)$.

**Proposition 3.** *For any given $\kappa \in (0,1)$, $0 < \epsilon < \kappa/2$, let $p_{min}$ denote the minimum propensity under supported set $p_{min} = max_{x,y \in \mathcal{U}(x,\pi_0)^c} \pi_0(y|x)$, then with probability larger than $1 - 2\exp(-2n\epsilon^2 p_{min}^2)$, the constraint $1 - \kappa + \epsilon \leq S_{\mathcal{D}}(\pi|\pi_0) \leq 1 - \epsilon$ will ensure $0 \leq \mathcal{D}_{\mathcal{X}}(\pi|\pi_0) \leq \kappa$.*

We can thus use $1 - S_{\mathcal{D}}(\pi|\pi_0)$ as a surrogate for $\mathcal{D}_{\mathcal{X}}(\pi|\pi_0)$ in the training objective:

$$\underset{\pi_w \in \Pi}{\arg\min} \frac{1}{n}\sum_{i=1}^{n} \frac{\pi_w(y_i|x_i)}{\pi_0(y_i|x_i)} r_i. \text{ subject to } 1 - \kappa + \epsilon \leq \frac{1}{n}\sum_{i=1}^{n} \frac{\pi_w(y_i|x_i)}{\pi_0(y_i|x_i)} \leq 1 - \epsilon \tag{11}$$

Using Lagrange multipliers, an equivalent dual form of Equation (11) is:

$$\underset{u_1, u_2 \geq 0}{\max} \underset{\pi_w \in \Pi}{\min} \frac{1}{n}\sum_{i=1}^{n} \frac{\pi_w(y_i|x_i)}{\pi_0(y_i|x_i)}(r_i + u_1 - u_2) - u_1(1-\epsilon) + u_2(1 - \kappa + \epsilon) \tag{12}$$

For each fixed $(u_1, u_2)$ pair, the inner minimization objective is ERM with IPS under a shift of the reward. Instead of maximizing over $(u_1, u_2)$ in the outer objective, we treat $(u_1 - u_2)$ as a hyperparameter that we select on a validation set. We explore various estimators for this model-selection problem in Section 4.

Note that, among the methods we proposed for dealing with support deficiency, this approach is the most efficient to implement, and it does not require access to the logging policy during training or testing. Furthermore, the form of the inner objective coincides with that of BanditNet (Joachims et al., 2018), which is known to work well for deep network training by controlling propensity overfitting (Swaminathan & Joachims, 2015a).

## 4 EMPIRICAL EVALUATION

We empirically evaluate the effectiveness and robustness of the three proposed approaches: restricting the action space, conservative and regression extrapolation, as well as restricting the policy space. The semi-synthetic experiments are based on two real-world datasets: one is the popular image classification dataset CIFAR10 (Krizhevsky et al.) and the other is the credit-card fraud dataset of Dal Pozzolo et al. (2015). We use the naive IPS estimator and the regression-based Direct Method (DM) as baselines.

The experiments are set up as follows. We first create a train-validation-test split for both datasets. The training set is used to generate bandit datasets for learning, the validation set is used to generate bandit datasets for model selection, and the full-information test set serves as ground truth for evaluating the learned policies. To simulate bandit feedback for the CIFAR10 dataset, our experiment setup follows traditional supervised $\rightarrow$ bandit conversion for multi-class classification datasets (Beygelzimer & Langford, 2009). To not only have bandit data with binary multi-class rewards, we

choose a different methodology for the credit-card dataset by designating some features as corresponding to actions and rewards. More details are given in Appendix B.

To get logging policies for generating bandit feedback, we start by training a softmax-policy as in Equation (4) on a subset of the full-information data. We then introduce a temperature parameter $\tau$ into the learned policy via $\tau f_w(x, y)$ to be able to control its stochasticity and support deficiency. In particular, we enforce zero support for some actions by clipping the propensities to 0 if they are below a threshold of $\epsilon = 0.01$. The larger $\tau$, the higher the support deficiency. Note that making the threshold at $\epsilon = 0.01$ allows us to control support while the variance of IPS stays bounded. This allows us to study support deficiency without having to worry about variance control.

For both logging and target policies, we train softmax policies where $f_w(x, y)$ is a neural network. We use the ResNet20 architecture (He et al., 2016) for CIFAR10, and a fully-connected 2-layer network for the credit-card dataset.

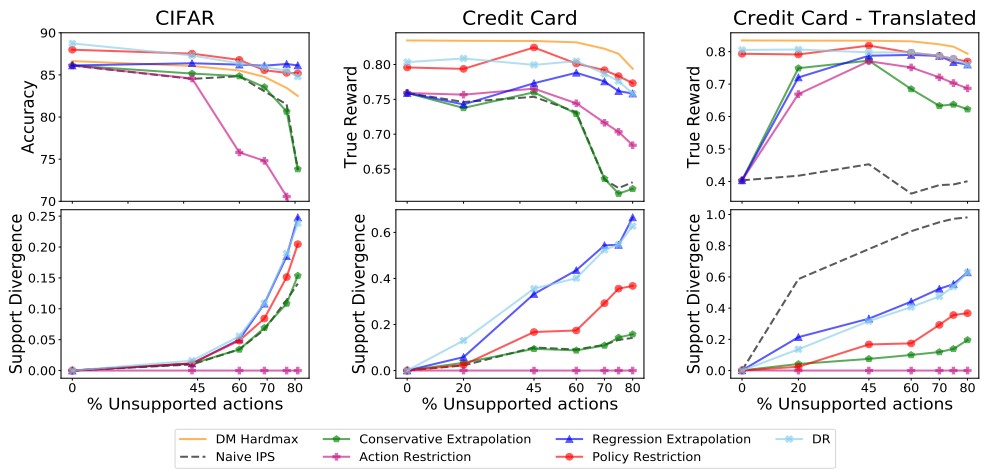

Figure 1: Learning results with varying support deficiency in the logging policy.

**How do the methods perform at different level of support deficiency?** Results are shown in Figure 1. First, as expected, learning using naive IPS degrades on both datasets as we make the logging policy more peaked and the number of unsupported actions increases. Note that naive IPS coincides with Conservative Extrapolation, since both datasets are scaled to have a minimum reward of zero. In the rightmost column, however, we translated the rewards to $[-1, 0]$. This has a strong detrimental effect on naive IPS, as it is now overly optimistic about unsupported actions. Second, the approach of dealing with support deficiency by restricting the action space also performs poorly. The second row of plot sheds some light on this, as it shows the support divergence $\mathcal{D}_{\mathcal{X}}(\pi|\pi_0)$ of the learned policy. It is zero for Action Restriction as expected, which means that bias is not the problem. Instead, as the number of unsupported actions increases, the best actions are more likely to be pruned and unavailable in the restricted policy space $\bar{\Pi}$. Third, Regression Extrapolation performs better than Conservative Extrapolation on both datasets. In both cases, the DM model is quite good which also benefits Regression Extrapolation. However, on the credit-card dataset the regression seems better at ranking than at predicting the true reward, which explains why DM performs better than Regression Extrapolation. Fourth, the method that performs well most consistently is Policy Restriction. Unlike all the other IPS-based methods, it performs well even under the translated rewards in the third column of Figure 1. This is because the objective of Policy Restriction coincides with that of BanditNet (Joachims et al., 2018), which is known to remedy propensity overfitting due to the lack of equivariance of the IPS estimator (Swaminathan & Joachims, 2015b).

**How does the learning performance change with more training data?** Results are shown in Figure 2. As the number of bandit examples increases, Policy Restriction, Regression Extrapolation and DM dominate over most of the range especially when the percentage of unsupported actions is large. Among the other methods, Action Restriction can take the least advantage of more data. This is plausible, since its maximum performance is limited by the available actions. For similar reasons, Conservative Extrapolation (and equivalently IPS) also flattens out, since it also tightly restricts the action space by imputing the minimum reward.

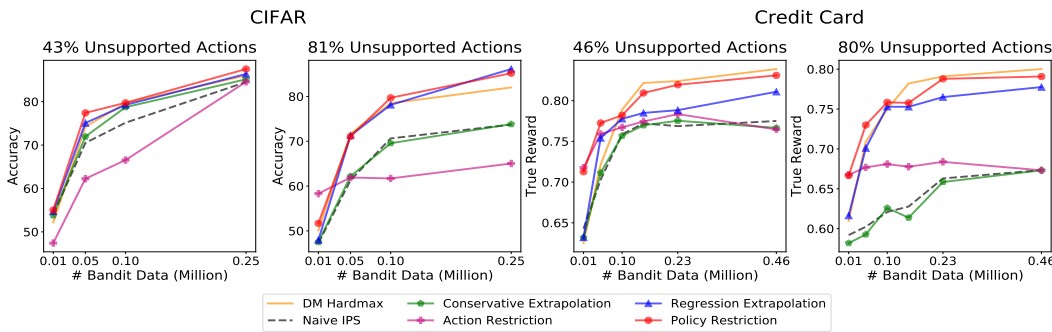

Figure 2: Learning results with varying amounts of bandit data on CIFAR10 and credit-card dataset.

| %
Unsupp. | Oracle | Regr.
Extrap. | DM | Cons.
Extrap. | SNIPS |
|---|---|---|---|---|---|
| 45 | 0.878 | **0.878** | **0.878** | **0.878** | 0.876 |
| 60 | 0.871 | **0.871** | **0.871** | **0.871** | **0.871** |
| 70 | 0.858 | **0.858** | 0.856 | **0.858** | **0.858** |
| 77 | 0.856 | 0.854 | 0.854 | **0.856** | **0.856** |
| 80 | 0.855 | **0.855** | **0.855** | 0.838 | 0.849 |

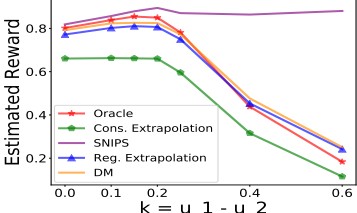

Figure 3: Model selection performance on CIFAR10.

**How effective are the estimators for model selection?** Most learning algorithms have hyperparameters, and we now evaluate how the estimators perform for this secondary learning problem. We specifically focus on the parameter $k := u_1 - u_2$ in Policy Restriction, since it controls how much the learned policies can step outside the region of support. The table on the left of Figure 3 shows the reward of the learned policy when performing model selection with the respective estimator. Oracle is the estimator that has access to the full-information validation set, and can thus be considered as a skyline. We also included the SNIPS estimator (Swaminathan & Joachims, 2015b), which imputes the average reward on the supported action for the unsupported actions (Gilotte et al., 2018). All estimators perform quite well for model selection on CIFAR, and the results are analogous for the credit-card data (see Appendix B.2). However, the plot to the right of Figure 3 reveals that SNIPS does not accurately reflect the shape of the Oracle curve. Both Regression Extrapolation and DM, however, are found to be sufficiently accurate for reliable model selection.

## 5 DISCUSSION AND CONCLUSIONS

We identified and analyzed how off-policy learning based on IPS weighting can suffer severely degraded learning performance when the logging policy is support deficient. To remedy this problem, we explored approaches that limit the impact of missing support through three different means: restricting the action space, reward extrapolation and restricting the policy space. We find that the most natural approach of restricting the action space is neither computationally efficient, nor does it learn accurate policies. Reward extrapolation through regression and restricting the policy space, however, both perform well and robustly even at high levels of support deficiency. Among those two methods, reward extrapolation has the potential drawback that we need to compute (and/or sample from) the complement of the logging policy, which can be computationally challenging. Furthermore, having to store all old logging policies is inconvenient in practice. This makes the approach of restricting the policy space particularly attractive, since it is computationally efficient and it does not require access to the logging policy beyond the logged propensity values.

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

# A  APPENDIX: PROOFS

In this appendix, we provide proofs of the main theorems and propositions.

## A.1  PROOF OF PROPOSITION 1

**Proposition 1.** *Given contexts $x \sim P(\mathcal{X})$ and logging policy $\pi_0(\mathcal{Y}|x)$, the bias of $\hat{R}_{IPS}$ for target policy $\pi(\mathcal{Y}|x)$ is equal to the expected reward on the unsupported action sets, i.e., $bias(\pi|\pi_0) = \mathbb{E}_x[-\sum_{y \in \mathcal{U}(x,\pi_0)} \pi(y|x)\delta(x,y)]$.*

*Proof.* Recall $\delta(x,y) = \mathbb{E}_r[r(x,y)|x,y]$, and logged data $\mathcal{D} \sim \mathcal{P}_{\mathcal{X}} \times \pi_0(\cdot|\mathcal{X}) \times \mathcal{P}_r$.

$$
\begin{aligned}
bias(\pi|\pi_0) &= \mathbb{E}_{\mathcal{D}}[\hat{R}_{IPS}(\pi)] - R(\pi) \\
&= \mathbb{E}_x[\sum_{y \in (\mathcal{U}(x,\pi_0))^c} \pi_0(y|x)\frac{\pi(y|x)}{\pi_0(y|x)}\delta(x,y) - \sum_{y \in \mathcal{Y}} \pi(y|x)\delta(x,y)] \\
&= \mathbb{E}_x[-\sum_{y \in \mathcal{U}(x,\pi_0)} \pi(y|x)\delta(x,y)]
\end{aligned}
\tag{13}
$$

$\square$

## A.2  PROOF OF THEOREM 1

**Theorem 1.** *For any given hypothesis space $\Pi$ with logging policy $\pi_0 \in \Pi$, there exists a reward distribution $\mathcal{P}_r$ with support in $[r_{min}, r_{max}]$ such that in the limit of infinite training data, ERM using IPS over the logged data $\mathcal{D} \sim P(\mathcal{X}) \times \pi_0(\cdot|\mathcal{X}) \times \mathcal{P}_r$ can select a policy $\hat{\pi} \in \arg\max_{\pi \in \Pi} \mathbb{E}_{\mathcal{D}}[\hat{R}_{IPS}(\pi)]$ that is at least $(r_{max} - r_{min}) \max_{\pi \in \Pi} \mathcal{D}_{\mathcal{X}}(\pi|\pi_0)$ suboptimal.*

*Proof.* For any given hypothesis space $\Pi$ and logging policy $\pi_0$, define a deterministic reward distribution $\mathcal{P}_r$ as the following: for all context $x$, $r(x,y) = \delta(x,y) = r_{min}$ for $y \in \mathcal{U}(x,\pi_0)^c$ and $r(x,y) = \delta(x,y) = r_{max}$ for $y \in \mathcal{U}(x,\pi_0)$. Let $\tilde{\pi} \in \arg\max_{\pi \in \Pi} \mathcal{D}_{\mathcal{X}}(\pi|\pi_0)$ and $\pi^* \in \arg\max_{\pi \in \Pi} R(\pi)$, then we have the following lower bound for $R(\pi^*)$:

$$
\begin{aligned}
R(\pi^*) &\geq R(\tilde{\pi}) \\
&= \mathbb{E}_x[\sum_{y \in \mathcal{U}(x,\pi_0)} r_{max} + \sum_{y \in \mathcal{U}(x,\pi_0)^c} r_{min}] \\
&= r_{max} \max_{\pi \in \Pi} \mathcal{D}_{\mathcal{X}}(\pi|\pi_0) + r_{min}(1 - \max_{\pi \in \Pi} \mathcal{D}_{\mathcal{X}}(\pi|\pi_0))
\end{aligned}
\tag{14}
$$

where the first inequality follows from the definition of $\pi^*$, the first and second equality is based on the specific reward distribution $\mathcal{P}_r$ and the definition of $\tilde{\pi}$.

In the following we will show that for any $\hat{\pi}$ learned by the expectation of ERM (or in the limit of infinite amount data), i.e., $\hat{\pi} \in \arg\max \mathbb{E}_{\mathcal{D}}[\hat{R}_{IPS}^{\mathcal{D}}(\pi)]$, $\hat{\pi}$ have the same support as $\pi_0$.

$$
\mathbb{E}_{\mathcal{D}}[\hat{R}_{IPS}(\pi)] = \mathbb{E}_x[\sum_{y \in \mathcal{U}(x,\pi_0)^c} \pi(y|x)r_{min}] = r_{min}\mathbb{E}_x[\sum_{y \in \mathcal{U}(x,\pi_0)^c} \pi(y|x)] \leq r_{min}
\tag{15}
$$

for all $\pi \in \Pi$, then it is easy to see $\pi_0 \in \Pi$ is one of the solution of ERM. Actually for any $\hat{\pi} \in \arg\max \mathbb{E}_{\mathcal{D}}[\hat{R}_{IPS}^{\mathcal{D}}(\pi)]$, $\mathbb{E}_x[\sum_{y \in \mathcal{U}(x,\pi_0)^c} \pi(y|x)] = 1$ and it gives us that any solution of the ERM has exactly the same support as $\pi_0$, then we have $R(\hat{\pi}) = r_{min}$ for $\hat{\pi} \in \arg\max \mathbb{E}_{\mathcal{D}}[\hat{R}_{IPS}^{\mathcal{D}}(\pi)]$.

Combining the lower bound for $R(\pi^*)$ and $R(\hat{\pi}) = r_{min}$, we have

$$
\begin{aligned}
R(\pi^*) - R(\hat{\pi}) &\geq r_{max} \max_{\pi \in \Pi} \mathcal{D}_{\mathcal{X}}(\pi|\pi_0) + r_{min}(1 - \max_{\pi \in \Pi} \mathcal{D}_{\mathcal{X}}(\pi|\pi_0)) - r_{min} \\
&= (r_{max} - r_{min}) \max_{\pi \in \Pi} \mathcal{D}_{\mathcal{X}}(\pi|\pi_0)
\end{aligned}
\tag{16}
$$

$\square$

### A.3 PROOF OF THEOREM 2

**Theorem 2.** *For contexts $x_i$ drawn i.i.d from $P(\mathcal{X})$, action $y_i$ drawn from logging policy $\pi_0$, we define $S_{\mathcal{D}}(\pi|\pi_0) = \frac{1}{n}\sum_{i=1}^{n} \frac{\pi(y_i|x_i)}{\pi_0(y_i|x_i)}$. For any policy $\pi$ it holds that*

$$\underset{x \sim P(\mathcal{X})}{\mathbb{E}} \underset{y \sim \pi_0(\cdot|x)}{\mathbb{E}} [S_{\mathcal{D}}(\pi|\pi_0)] + \mathcal{D}_{\mathcal{X}}(\pi|\pi_0) = 1 \qquad (10)$$

*Proof.*

$$
\begin{aligned}
\underset{x,y \sim \pi_0}{\mathbb{E}} [S_{\mathcal{D}}(\pi|\pi_0)] + \mathcal{D}_{\mathcal{X}}(\pi|\pi_0) &= \underset{x}{\mathbb{E}}[\sum_{y \in \mathcal{U}(x,\pi_0)^c} \pi_0(y|x)\frac{\pi(y|x)}{\pi_0(y|x)}] + \mathcal{D}_{\mathcal{X}}(\pi|\pi_0) \\
&= \underset{x}{\mathbb{E}}[\sum_{y \in \mathcal{U}(x,\pi_0)^c} \pi(y|x)] + \underset{x}{\mathbb{E}}[\sum_{y \in \mathcal{U}(x,\pi_0)} \pi(y|x)] \qquad (17) \\
&= \underset{x}{\mathbb{E}}[\sum_{y \in \mathcal{Y}} \pi(y|x)] = 1
\end{aligned}
$$

The first equality is based on definition of $S_{\mathcal{D}}(\pi|\pi_0)$ and the second equality is based on definition of support divergence. □

### A.4 PROOF OF PROPOSITION 3

**Proposition 3.** *For any given $\kappa \in (0,1)$, $0 < \epsilon < \kappa/2$, let $p_{min}$ denote the minimum propensity under supported set $p_{min} = max_{x,y \in \mathcal{U}(x,\pi_0)^c} \pi_0(y|x)$, then with probability larger than $1 - 2\exp(-2n\epsilon^2 p_{min}^2)$, the constraint $1 - \kappa + \epsilon \leq S_{\mathcal{D}}(\pi|\pi_0) \leq 1 - \epsilon$ will ensure $0 \leq \mathcal{D}_{\mathcal{X}}(\pi|\pi_0) \leq \kappa$.*

*Proof.* Recall $S_{\mathcal{D}}(\pi|\pi_0) = \frac{1}{n}\sum_{i=1}^{n} \frac{\pi(y_i|x_i)}{\pi_0(y_i|x_i)}$ with $(x_i, y_i)$ draw i.i.d from $P(\mathcal{X}) \times \pi_0(\mathcal{Y}|x)$. From Appendix A.3, it is easy to see $\mathbb{E}_{x,y \sim \pi_0(\cdot|x)}[\frac{\pi(y|x)}{\pi_0(y|x)}] = 1 - \mathcal{D}_{\mathcal{X}}(\pi|\pi_0)$. Let $p_{min}$ denote the smallest propensity under supported action set, $p_{min} := min_{x,y \in \mathcal{U}(x,\pi_0)^c} \pi_0(y|x) > 0$, then the random variable $\frac{\pi(y|x)}{\pi_0(y|x)}$ is strictly bounded between $[0, \frac{1}{p_{min}}]$. Applying Hoeffding's bound gives:

$$\mathbb{P}(\mathcal{D}_{\mathcal{X}}(\pi|\pi_0) < 1 - S_{\mathcal{D}}(\pi|\pi_0) - \epsilon) = \mathbb{P}(S_{\mathcal{D}}(\pi|\pi_0) - (1 - \mathcal{D}_{\mathcal{X}}(\pi|\pi_0)) < -\epsilon) \leq exp(-2n\epsilon^2 p_{min}^2) \qquad (18)$$

Since $S_{\mathcal{D}(\pi|\pi_0)} \leq 1 - \epsilon$ gives $1 - S_{\mathcal{D}}(\pi|\pi_0) - \epsilon \geq 0$, then we have

$$\mathbb{P}(\mathcal{D}_{\mathcal{X}}(\pi|\pi_0) < 0) \leq exp(-2n\epsilon^2 p_{min}^2) \qquad (19)$$

Similar for the other direction, Hoeffding's bound gives:

$$\mathbb{P}(\mathcal{D}_{\mathcal{X}}(\pi|\pi_0) > 1 - S_{\mathcal{D}}(\pi|\pi_0) + \epsilon) = \mathbb{P}(S_{\mathcal{D}}(\pi|\pi_0) - (1 - \mathcal{D}_{\mathcal{X}}(\pi|\pi_0)) > \epsilon) \leq exp(-2n\epsilon^2 p_{min}^2) \qquad (20)$$

Since $S_{\mathcal{D}(\pi|\pi_0)} \geq 1 + \epsilon - \kappa$ gives $1 - S_{\mathcal{D}}(\pi|\pi_0) + \epsilon \leq \kappa$, then we have

$$\mathbb{P}(\mathcal{D}_{\mathcal{X}}(\pi|\pi_0) \geq \kappa) \leq exp(-2n\epsilon^2 p_{min}^2) \qquad (21)$$

Combining the above, we have

$$
\begin{aligned}
\mathbb{P}(0 \leq \mathcal{D}_{\mathcal{X}}(\pi|\pi_0) \leq \kappa) &= 1 - \mathbb{P}(\mathcal{D}_{\mathcal{X}}(\pi|\pi_0) < 0) - \mathbb{P}(\mathcal{D}_{\mathcal{X}}(\pi|\pi_0) > \kappa) \\
&\geq 1 - 2exp(-2n\epsilon^2 p_{min}^2)
\end{aligned} \qquad (22)
$$

□

### A.5 PROOF FOR EFFICIENT APPROXIMATION

**Claim 1.** *The empirical risk defined by in Equation (8) has the same expectation (over randomness in $\mathcal{D}$ and sampling) as $\hat{R}_{IPS}^{\delta}(\mathcal{D})$.*

*Proof.* Taking the expectation of empirical risk defined in Equation (8):

$$
\mathbb{E}\Big[\frac{1}{n}\sum_{i=1}^{n}\frac{\pi(y_i|x_i)}{\pi_0(y_i|x_i)}r_i + \frac{1}{m}\sum_{j=1}^{m}\frac{\pi(y_j|x_j)}{p_j}\hat{\delta}(x_j,y_j)\Big]
$$

$$
= \mathbb{E}_{x}\Big[\sum_{y\in\mathcal{U}(x,\pi_0)^c}\pi_0(y|x)\frac{\pi(y|x)}{\pi_0(y|x)}\delta(x,y)\Big] + \mathbb{E}_{x}\Big[\sum_{y\in\mathcal{U}(x,\pi_0)}\frac{1}{|\mathcal{U}(x,\pi_0)|}\frac{\pi(y|x)}{\frac{1}{|\mathcal{U}(x,\pi_0)|}}\hat{\delta}(x,y)\Big] \tag{23}
$$

$$
= \mathbb{E}_{x}\Big[\sum_{y\in\mathcal{U}(x,\pi_0)^c}\pi(y|x)\delta(x,y)\Big] + \mathbb{E}_{x}\Big[\sum_{y\in\mathcal{U}(x,\pi_0)}\pi(y|x)\hat{\delta}(x,y)\Big]
$$

Now we will show it has the same expectation with $\hat{R}^{\delta}_{IPS}(\pi)$

$$
\mathbb{E}_{\mathcal{D}}\Big[\frac{\pi(y_i|x_i)}{\pi_0(y_i|x_i)}r_i + \sum_{y\in\mathcal{U}(x_i,\pi_0)}\pi(y|x_i)\hat{\delta}(x_i,y)\Big]
$$

$$
= \mathbb{E}_{x}\Big[\mathbb{E}_{y\sim\pi_0}[\frac{\pi(y|x)}{\pi_0(y|x)}\delta(x,y)] + \sum_{y'\in\mathcal{U}(x,\pi_0)}\pi(y'|x)\hat{\delta}(x,y')\Big]
$$

$$
= \mathbb{E}_{x}[\sum_{y\in\mathcal{U}(x,\pi_0)^c}\pi_0(y|x)\frac{\pi(y|x)}{\pi_0(y|x)}\delta(x,y) + \sum_{y'\in\mathcal{U}(x,\pi_0)}\pi(y'|x)\hat{\delta}(x,y')] \tag{24}
$$

$$
= \mathbb{E}_{x}\Big[\sum_{y\in\mathcal{U}(x,\pi_0)^c}\pi(y|x)\delta(x,y)\Big] + \mathbb{E}_{x}\Big[\sum_{y\in\mathcal{U}(x,\pi_0)}\pi(y|x)\hat{\delta}(x,y)\Big]
$$

The proof is done by comparing Equation (23) and Equation (24). □

## A.6 PROOF OF PROPOSITION 2

**Proposition 2.** *Given contexts $x_1, x_2, \ldots, x_n$ drawn i.i.d from the unknown distribution $P(\mathcal{X})$, for action $y_i$ drawn independently from logging policy $\pi_0$ with probability $\pi_0(\mathcal{Y}|x_i)$, the bias of the empirical risk defined in Equation (7) is $\mathbb{E}_x[\sum_{y\in\mathcal{U}_x^{\pi_0}}\pi(y|x)\Delta(x,y)]$.*

*Proof.* From Appendix A.5, we are given the expectation of $\hat{R}^{\delta}_{IPS}(\pi)$, and the bias is:

$$
bias(\hat{R}^{\delta}_{IPS}(\pi)) = \mathbb{E}_{\mathcal{D}}[\hat{R}^{\delta}_{IPS}(\pi)] - R(\pi)
$$

$$
= \mathbb{E}_{x}\Big[\sum_{y\in\mathcal{U}(x,\pi_0)^c}\pi(y|x)\delta(x,y) + \sum_{y\in\mathcal{U}(x,\pi_0)}\pi(y|x)\hat{\delta}(x,y)\Big] - R(\pi)
$$

$$
= \mathbb{E}_{x}\Big[\sum_{y\in\mathcal{U}(x,\pi_0)}\pi(y|x)(\hat{\delta}(x,y)-\delta(x,y))\Big] \tag{25}
$$

$$
= \mathbb{E}_{x}\Big[\sum_{y\in\mathcal{U}(x,\pi_0)}\pi(y|x)\Delta(x,y)\Big]
$$

The second equality is from Appendix A.5, the second equality is based on $R(\pi) = \mathbb{E}_x\Big[\sum_{y\in\mathcal{U}(x,\pi_0)^c}\pi(y|x)\delta(x,y) + \sum_{y\in\mathcal{U}(x,\pi_0)}\pi(y|x)\delta(x,y)\Big]$, and the last one is based on the definition of $\Delta(x,y) := \hat{\delta}(x,y) - \delta(x,y)$ for all $x\in\mathcal{X}, y\in\mathcal{Y}$. □

## B  APPENDIX: EXPERIMENTS

In this section, we provide the experiment details and additional results to help promote reproducibility of this work.

### B.1  EXPERIMENT SETUP DETAILS

**Datasets and baseline.**  We follow a 75:10:15 train-validation-test split for credit card fraud detection dataset, while for CIFAR10 already coming with a train-test split, we keep 10% of the training set as validation set. Baseline estimators are IPS and DM, the hyperparameters (learning rate, L2 regularization) are optimized for all the methods based on the validation set.

**Bandit data generation.**  For CIFAR10, given supervised data $\{x_i, y_i^*\}_{i=1}^n$ where $x_i$ denotes the 3072 features and $y_i^*$ denotes the correct label of data (ranging from 0 to 9), under logging policy $\pi_0$, the logged bandit data is generated by drawing $y_i \sim \pi_0(\mathcal{Y}|x_i)$, then a deterministic reward is defined as $\mathbb{I}_{\{y_i=y_i^*\}}$. For the credit card fraud detection dataset, we throw away the class label and only use the features for each sample to generate bandit data. To be specific, for each sample with a 28-dimensional feature vector, we define the first 20 features as the contextual information, and use the remaining 8 features as the underlying true reward for 8 different actions (with normalization).

**Logging policy.**  For CIFAR, we learn the softmax logging policy on 35K full-information data points as a multi-class classification problem with cross-entropy loss. Similar as the experiments on BanditNet (Joachims et al., 2018), we adopt the conventional ResNet20 architecture but restrict training after a mere two epochs to derive a relative stochastic policy, since it will be easier to add temperature later to control its stochasticity and support deficiency. Similarly, for the credit card fraud detection dataset, the softmax logging policy is learned on 8K full-information data points by treating it as a multi-class classification problem using cross-entropy loss and the label being the action with the highest reward on this specific context. For CIFAR, the logging policy we trained has a 57.43% accuracy on the test-set; whereas for the credit card fraud detection dataset, the logging policy has an expected true reward of 0.71.

**Reward estimator.**  For each experiment, we train a different regression function using the full bandit dataset. We use the same architecture as the one used for off-policy learning - where the final layer is the size of the actions, specifying the reward for each action given a particular context. The regression function is trained using the MSE objective.

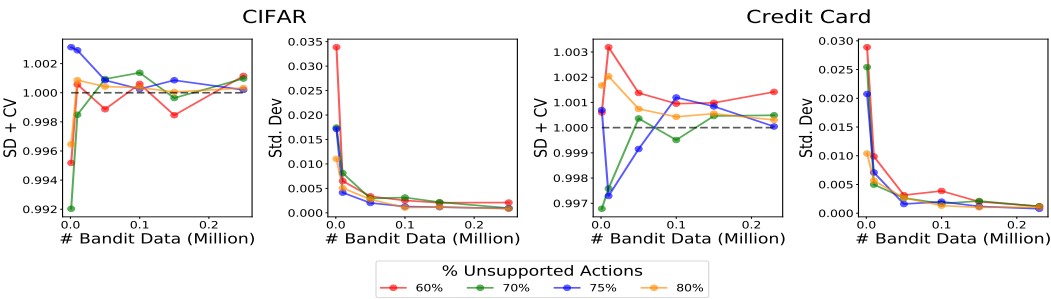

Figure 4: Behaviour of $S_{\mathcal{D}}(\pi|\pi_0) + \mathcal{D}_{\mathcal{X}}(\pi|\pi_0)$

### B.2  SUPPLEMENTARY RESULTS

**Reliability of approximating support divergence using control variate.**  In this experiment, we empirically verify the reliability of estimating support divergence using control variate. The target policy is the uniform policy while the logging policies are varying in their support deficiency. Results are averaged over 10 runs and is shown in Figure 4. We investigate the behaviour of $S_{\mathcal{D}}(\pi|\pi_0) + \mathcal{D}_{\mathcal{X}}(\pi|\pi_0)$ under different number of training data and different support deficiency of the corresponding logging policy. As we can see, the sum converges to 1 as the training data

| % Unsupp. | Oracle | Reg. Extrap. | DM | Cons. Extrap. | SNIPS |
|---|---|---|---|---|---|
| 0 | 0.7934 | **0.793** | 0.793 | **0.793** | 0.793 |
| 20 | 0.803 | 0.791 | **0.803** | 0.791 | **0.803** |
| 45 | 0.799 | 0.798 | **0.799** | 0.795 | 0.798 |
| 60 | 0.797 | **0.797** | 0.796 | **0.797** | 0.757 |
| 70 | 0.787 | **0.787** | 0.780 | 0.771 | 0.754 |
| 75 | 0.778 | **0.778** | **0.778** | 0.766 | 0.774 |
| 80 | 0.774 | 0.759 | 0.759 | 0.751 | 0.759 |

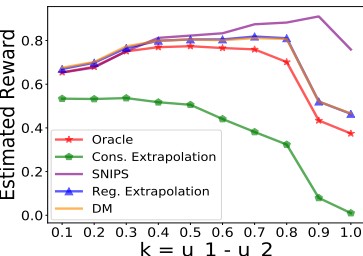

Figure 5: Model selection result for credit card fraud detection

increases. Meanwhile, the variance decreases as shown in the right figure. For different support deficiency curves, the curve converges in a similar fashion and we conjecture it is due to the effect of clipping the propensity at the same threshold $\epsilon = 0.01$, which makes $p_{min} = 0.01$ in the bound shown in Proposition 3.

**Model selection comparison for the credit card dataset.** The model selection comparison over the credit card fraud detection dataset is demonstrated in Figure 5. Similar as the trend in Figure 3, SNIPS and Conservative Extrapolation exhibit a large bias, also SNIPS even can not reflect the shape of the Oracle curve. DM and Regression Extrapolation closely track the Oracle line, and they have the best performance when used in model selection, as seen in the left table of Figure 5.

