# OpenReview forum: "Off-policy Bandits with Deficient Support"
_ICLR.cc/2020/Conference — Reject_

### Official Review · AnonReviewer3 · 2019-10-21
**Official Blind Review #3**

**Rating:** 3

**Review:**

This paper talks about the problem of off-policy or batch learning in the contextual bandit setting without the complete support assumption. This problem setting is very realistic and encountered in most problems, especially in temporally extended settings, such as reinforcement learning. They compare three approaches for the same: restricting action selection, learning extrapolated reward models, and by restricting the policy class. They derive a SNIPS style estimator for the support constraint in the final approach. The approach with restricting the policy class demonstrates decent empirical results although the direct method is very much comparable.

I would lean towards being mostly neutral in terms of acceptance. While the problem being solved is very relevant and their approach compares three different approaches to the deficient support problem, I am not sure how this work is positioned with respect to approaches solving similar problems in the reinforcement learning land. For example, Batch constrained Q-learning ([1]) restricts the set of actions that can be used, Bootstrapping Error Accumulation ([2]) and SPIBB ([3]) restrict the policy class in batch reinforcement learning. I would appreciate some comparison/positioning to such methods in the bandit setting as well. The support estimation metric and the corresponding objective (Eqn 10, 11) should also be compared and contrast with explicit divergences designed for support matching (for example, in [4]).

References:
[1] Off-Policy Deep reinforcement learning without exploration, Fujimoto et.al. ICML 2019
[2] Stabilizing Off-policy Q-learning via Bootstrapping Error Reduction, Kumar et.al. NeuRIPS 2019
[3] Safe Policy Improvement with Baseline Bootstrapping. Laroche et.al. ICML 2019
[4] Domain adaptation with asymmetrically-relaxed distribution alignment. Wu et.al. ICML 2019

**Experience Assessment:**

I have published one or two papers in this area.

**Review Assessment: Checking Correctness Of Derivations And Theory:**

I assessed the sensibility of the derivations and theory.

**Review Assessment: Checking Correctness Of Experiments:**

I assessed the sensibility of the experiments.

**Review Assessment: Thoroughness In Paper Reading:**

I read the paper at least twice and used my best judgement in assessing the paper.

---

> ### Author Response · Authors · 2019-11-13
> **Official Comment to Review #3**
>
> We thank the reviewer to point out the related work in sequential RL. We discuss the this work in more detail below, and will add a discussion to the final version of the paper. However, it is important be clear that these works address problems that primarily affect the sequential setting, not the contextual-bandit setting we study. In particular, the key problem of off-policy RL lies in the fact that importance-sampling based estimators suffers variance that grows exponentially with the horizon, making variance control (not bias control due to deficient support) the key problem. This is fundamentally different for one-step contextual bandits, where importance-sampling based estimator are the most competitive ones -- as can be clearly seen in our experiments. Meanwhile, Sutton and Barto (2018) identify a deadly triad of function approximation, bootstrapping, and off-policy learning, which emphasizes that function approximation equipped with Q-learning can even diverge in the off-policy learning setting, which makes most off-policy RL works tend to be very conservative in extrapolation error since the severe error propagation issue. In our work, we focus on the contextual bandit problem, which has many applications in recommender systems, ad placement and online search. We addresses different approaches to handle the important issue of support deficiency and empirically show the strong performance of safe learning by restricting policy class. The algorithm is simple and can be implemented efficiently using SGD. For practitioners, it has important usage in real world applications and for developing better off-policy estimators in RL, and we believe it is important to understand how to handle these cases in the more tractable contextual-bandit case.
>
> Discussing the specific paper you mention, SPIBB ([3]) restricts the policy class in the following way: when there is not enough data for a particular $(x,y)$ pair, it forces the policy to obey $\pi(y|x)=\pi_0(y|x)$, and all the remaining probability mass will be given to the greedy action which has the highest reward based on the reward model. This strategy heavily relies on having the same context appear both at learning and at prediction time, which is an unrealistic requirement in most contextual bandit applications (e.g. recommendation, advertisement). If the context $x$ did not appear during learning, then the learned policy will exactly mimic the logging policy, which leads to no improvement at all. To make experiments feasible, the author proposed a pseudo-count workaround based on the Euclidean state-distance to heuristically avoid this strict requirement, bringing the problem back to a form of (difficult to analyze) extrapolation.
>
> For [2], when translated into the bandit setting, the training objective will be the average reward predicted by the model ensembles, with a variance penalty and a constraint on the policy class. The constraint is based on the MMD between the logging policy and the target policy, and they argue heuristically that MMD is a good distance to measure support mismatch. Compared to our proposed approach of safe learning through constraining the policy space, we directly control the support divergence by using the control variate as a surrogate, which is more explicit and theoretically sound divergence measure. Moreover, their objective is based on an ensemble of direct models, while ours improved upon importance based methods. We want to emphasize that there is no end to improving the quality of regression models and people can use any deep learning architecture, however the bias problem could be severe if a wrong model is chosen, and the high bias problem is very difficult to diagnostic in general. While our recommended approach (safe learning by restricting policy class) is more safe in the sense of not relying on this extrapolated models, and then directly control the support divergence between two policies.
>
> BCQ [1] deals with batch RL with continuous actions. When simplified and translated to contextual bandits with discrete actions, it basically has a generative model to generate similar actions (compared with the actions shown in the batch) for each context. It then selects the sampled action that has the highest estimated reward. This is equivalent to a direct modeling approach in an action space that is restricted by the sampling procedure. As the sample size increases, this sampled action space converges to the support set of the logging policy. Similar to [2], it may suffer from high bias of the regression model, and it heavily relies on how good the reward estimate is.
>
> References:
>
> [1] Off-Policy Deep reinforcement learning without exploration, Fujimoto et.al. ICML 2019
> [2] Stabilizing Off-policy Q-learning via Bootstrapping Error Reduction, Kumar et.al. NeuRIPS 2019
> [3] Safe Policy Improvement with Baseline Bootstrapping. Laroche et.al. ICML 2019

---

### Official Review · AnonReviewer2 · 2019-10-23
**Official Blind Review #2**

**Rating:** 3

**Review:**

This paper considers a new off-policy contextual-bandit method that can learn even when the logging policy has deficient support. Three approaches are explored, namely restricting the action space, reward extrapolation, and restricting the policy space.

This paper is well written and it considers an important problem of deficient support. However, the proposed method was only compared to a few old benchmarks. How does the proposed method compare to more recent state-of-the-art off-policy bandit approaches (Liu et al. (2019), Xie et al. (2019), Tang et al. (2019)) in the experiments? The work by Liu et al. (2019) also considered the setting of deficient support.

Yao Liu, Adith Swaminathan, Alekh Agarwal, and Emma Brunskill. Off-policy policy gradient with state distribution correction. arXiv:1904.08473, 2019.

Jie, Liu, Liu, Wang, and Peng, Off-Policy Evaluation and Learning from Logged Bandit Feedback: Error Reduction via Surrogate Policy. ICLR 2019.

Tang, Feng, Li, Zhou, and Liu, Doubly Robust Bias Reduction in Infinite Horizon Off-Policy Estimation, arxiv: 1910.07186, 2019.




**Experience Assessment:**

I have read many papers in this area.

**Review Assessment: Checking Correctness Of Derivations And Theory:**

I assessed the sensibility of the derivations and theory.

**Review Assessment: Checking Correctness Of Experiments:**

I assessed the sensibility of the experiments.

**Review Assessment: Thoroughness In Paper Reading:**

I made a quick assessment of this paper.

---

> ### Author Response · Authors · 2019-11-13
> **Official Comment to Review #2**
>
> We thank the reviewer for pointing out the related papers handling off-policy learning in contextual bandit problems. Liu's work, as we mentioned in the related work section, focuses on the correction of the state distribution by defining an augmented MDP, and pessimistic imputation is used to get an estimate for policy-gradient learning. When we translate this pessimistic imputation idea into the contextual bandit setting, it is the same as the Conservative Extrapolation method we define in Section 3.2. We will make this relationship more explicit in the revised version of the paper. Note that we do provide experiments for Conservative Exploration (see Figure 1 and Figure 2), and it is clear that this method is too pessimistic and hence not recommended.
>
> As for Xie's work, it is based on the standard IPS estimator. However, it uses estimated propensities (based on maximum likelihood estimation) instead of the true propensities. When the logging policy has full support, using the estimated propensities can reduce the variance of IPS and its asymptotic MSE. However, it provides no remedy for the bias that IPS incurs under deficient support. In this paper, we emphasize that our goal is to address the bias problem of importance-sampling based estimators under deficient support, and IPS based on any surrogate policy still suffers from this severe bias problem.
>
> For Tang's work, it is based on the IH estimator proposed in RL, which is specifically designed to handle the unbounded variance problem of IPS under infinite horizon. Specifically, they propose a doubly robust approach to further reduce the bias of this estimator. When translating it into the bandit setting, it becomes the standard doubly robust estimator, and we argue in Section 3.2 that this is a special case of regression extrapolation when one changes the base estimator from IPS to DR (i.e. change the first term of Equation (7) to the corresponding DR estimator). We have added empirical results for DR in the updated version of the paper (See Figure 1). It turns out DR is no better than Policy Restriction, and the performance of DR is highly related to the performance of direct models, while our proposed Policy Restriction is model-free.
>
> Reference:
>
> Yao Liu, Adith Swaminathan, Alekh Agarwal, and Emma Brunskill. Off-policy policy gradient with state distribution correction. arXiv:1904.08473, 2019.
>
> Xie, Liu, Liu, Wang, and Peng, Off-Policy Evaluation and Learning from Logged Bandit Feedback: Error Reduction via Surrogate Policy. ICLR 2019.
>
> Tang, Feng, Li, Zhou, and Liu, Doubly Robust Bias Reduction in Infinite Horizon Off-Policy Estimation, arxiv: 1910.07186, 2019.

---

> > ### Comment · AnonReviewer2 · 2019-11-13
> > **new simulation**
> >
> > Thank the authors for adding the DR method in Tang et al. (2019) to Figure 1. It looks that the proposed three methods could not clearly beat this DR method in terms of accuracy. I understand that the DR is model-based and the proposed one is model-free. But it is not convincing if the newly proposed method could not beat the existing method in experiments.

---

> > > ### Author Response · Authors · 2019-11-15
> > > **Response**
> > >
> > > Thank you for the response. On the narrow question of DR vs. policy restriction, we do maintain that policy restriction is preferable, since it does not require training and optimizing a regression model.
> > >
> > > Stepping back and taking a broader view, the main contribution of the paper is not any particular method. Instead, it is the first paper to comprehensively investigate the problem of support deficiency in the contextual-bandit setting, which all reviewers agree is an important and pervasive problem in many real-world systems. In particular, it articulates that existing approaches can be structured into action restriction, regression imputation, and policy restriction, characterizes these three approaches theoretically and empirically, and then provides tangible recommendations that are far from obvious and valuable in practice. We suggest to evaluate the contribution of the paper along this broader view.

---

### Official Review · AnonReviewer1 · 2019-10-23
**Official Blind Review #1**

**Rating:** 6

**Review:**

This work addresses the problem of off-policy evaluation in the presence of positivity violations, i.e. some actions are not observed in the logged policy. As the paper points out, positivity violations can lead to unboundedly bad estimates when employing IPS. The authors propose three methods to deal with this problem. The first uses only the observed actions, the second and third use extrapolation and augmentation to provide an approximation to the off-policy problem.

I found a few pieces of this paper confusing. In section 3.2 it is proposed that a surrogate reward function be used for actions with unknown support, but the left hand side of the equation would seem to imply that the ratio still needs to be known in order to get an estimate. Perhaps an indicator function is missing?

It is also not made plain what assumptions are being employed in order to allow for extrapolation. From what I can tell, the authors are swapping out a positivity assumption with a smoothness assumption on the reward function. However, I don't think I see this spelled out within the text.

Overall, I think this is a promising approach (the empirical results certainly bare that outO but to my eyes it lacks sufficient detail and specificity.

**Experience Assessment:**

I have read many papers in this area.

**Review Assessment: Checking Correctness Of Derivations And Theory:**

I assessed the sensibility of the derivations and theory.

**Review Assessment: Checking Correctness Of Experiments:**

I carefully checked the experiments.

**Review Assessment: Thoroughness In Paper Reading:**

I read the paper thoroughly.

---

> ### Author Response · Authors · 2019-11-13
> **Official Comment to Review #1**
>
> We thank the reviewer for the comment and we clarify Section 3.2 as follows. In this section, we introduce safe learning through reward extrapolation, where the augmented IPS estimator listed in Equation (7) is composed of two parts: the first IPS part serves as an unbiased estimate for the reward of policy $\pi$ if limited to the actions which lie in the support of $\pi_0$, i.e., $E_{x}E_{y\notin \mathcal{U}(x,\pi_0)}[r(x,y)]$. The ratio (importance sampling weight) is fully known since we know the target policy $\pi$ and also the logging propensities $\pi_0(y_i|x_i)$.  The second imputation part is to extrapolate the reward for the actions that do not lie in the support of $\pi_0$, i.e. $E_{x}E_{y\in \mathcal{U}(x,\pi_0)}[r(x,y)]$, where the estimate for this part is based on a regression model $\hat{\delta}(x,y)$.
>
> As we mentioned in the paper, all regression-based approaches rely on a correctly specified parametric model, or on some smoothness assumption about the reward, in order to extrapolate well. We formally show the bias of this estimator in Proposition 2, and it is quantified by using the error of the regression model at the actions which do not lie in the support set of logging policy. If there exists prior knowledge on the exact form of the reward model, then reward extrapolation is optimal and recommended. However, this is typically not the case in most real world scenarios, where the bias of the reward model is typically unknown and could not be efficiently estimated. This is partly the reason why we advocate for using the method that restricts the policy space.

---

### Decision · Program_Chairs · 2019-12-19

**Decision:**

Reject

**Comment:**

This paper tackles the problem of learning off-policy in the contextual bandit problem, more specifically when the available data is deficient (in the sense that it does not allow to build reasonable counterfactual estimators). To address this, the authors introduce three strategies: 1) restricting the action space; 2) imputing missing rewards when lacking data; 3) restricting the policy space to policies with "enough" data. All three approaches are analyzed (statistical and computational properties) and evaluated empirically. Restricting the policy space appears to be particularly effective in practice.

Although the problem being solved is very relevant,  it is not clear how this work is positioned with respect to approaches solving similar problems in RL. For example, Batch constrained Q-learning ([1]) restricts action space, while Bootstrapping Error Accumulation ([2]) and SPIBB ([3]) restrict the policy class in batch RL. A comparison with these techniques in the contextual bandit settings, in addition to recent state-of-the-art off-policy bandit approaches (Liu et al. (2019), Xie et al. (2019)) is lacking. Moreover, given the newly added results (DR method by Tang et al. (2019)), it is not clear how the proposed approach improves over existing techniques. This should be clarified. I therefore recommend to reject this paper.